# Microglia and CD206+ border-associated mouse macrophages maintain their embryonic origin during Alzheimer's disease

Xiaoting Wu[1], Takashi Saito[2], Takaomi C Saido[3], Anna M Barron[4], Christiane Ruedl[1]*

[1]School of Biological Sciences, Nanyang Technological University, Singapore, Singapore; [2]Department of Neurocognitive Science, Institute of Brain Science, Nagoya City University Graduate School of Medical Sciences, Nagoya, Japan; [3]Laboratory for Proteolytic Neuroscience, RIKEN Center for Brain Science, Wako-Shi, Japan; [4]Lee Kong Chian School of Medicine, Nanyang Technological University, Singapore, Singapore

**Abstract** Brain microglia and border-associated macrophages (BAMs) display distinct spatial, developmental, and phenotypic features. Although at steady state, the origins of distinct brain macrophages are well-documented, the dynamics of their replenishment in neurodegenerative disorders remain elusive, particularly for activated CD11c+ microglia and BAMs. In this study, we conducted a comprehensive fate-mapping analysis of murine microglia and BAMs and their turnover kinetics during Alzheimer's disease (AD) progression. We used a novel inducible AD mouse model to investigate the contribution of bone marrow (BM) cells to the pool of fetal-derived brain macrophages during the development of AD. We demonstrated that microglia remain a remarkably stable embryonic-derived population even during the progression of AD pathology, indicating that neither parenchymal macrophage subpopulation originates from, nor is replenished by, BM-derived cells. At the border-associated brain regions, bona fide CD206+ BAMs are minimally replaced by BM-derived cells, and their turnover rates are not accelerated by AD. In contrast, all other myeloid cells are swiftly replenished by BM progenitors. This information further elucidates the turnover kinetics of these cells not only at steady state, but also in neurodegenerative diseases, which is crucial for identifying potential novel therapeutic targets.

*For correspondence:
ruedl@ntu.edu.sg

Competing interest: The authors declare that no competing interests exist.

## Introduction

Microglia are brain parenchymal macrophages that are unique among tissue-resident macrophages due to their primitive yolk sack origin, self-renewal properties (*Ajami et al., 2007*), and independence from adult hematopoiesis (*Ginhoux et al., 2010*; *Schulz et al., 2012*; *Sheng et al., 2015*). These cells are essential for normal neuronal development, brain function, and central nervous system (CNS) homeostasis, and dysfunction is associated with severe brain neuropathologies (*Leng and Edison, 2021*; *Sevenich, 2018*). In neurodegenerative diseases, such as Alzheimer's disease (AD), microglia form a barrier around amyloid plaques, thereby protecting against the neurotoxicity of aggregated beta-amyloid (Aβ) (*Condello et al., 2015*). Disruption of microglial activity, which is often related to aging and inflammation, can promote this neurotoxicity (*Salter and Stevens, 2017*) and affect the clearance of Aβ aggregates (*Floden and Combs, 2011*). The resulting increase in plaque formation (*Spangenberg et al., 2019*) leads to the progression of pathogenesis.

In several neurodegenerative diseases, the gradual appearance of a subpopulation of CD11c[+] activated microglia, also known as disease-associated microglia or neurodegenerative microglia, correlates with the progression of disease pathology (*Deczkowska et al., 2018*; *Keren-Shaul et al., 2017*; *Krasemann et al., 2017*). In AD, the accumulation of these highly phagocytic cells around the Aβ deposits (*Kamphuis et al., 2016*; *Yin et al., 2017*) mitigates Aβ-associated tau seeding and spreading (*Gratuze et al., 2021*). These activated microglia may play a protective role in neurodegeneration, thus implicating targeted manipulation as a therapeutic strategy. Although the origin of the myeloid cells that accumulate around the Aβ deposits is controversial, early studies indicated selective migration of monocytic cells toward the Aβ plaques (*Hohsfield and Humpel, 2015*), while more recent research suggests that these cells are derived from resident embryonic-derived microglia (*Reed-Geaghan et al., 2020*; *Shukla et al., 2019*).

In addition to the macrophages in brain parenchyma, microglia-independent macrophages are also found at the border regions, such as the subdural meninges (SDM), including the pia and arachnoid mater, the dura mater (DM), and the choroid plexus (CP) (*Brioschi et al., 2020*; *Utz et al., 2020*). Single-cell RNAseq analysis revealed that these border-associated macrophages (BAMs) represent a family of different subpopulations (*Van Hove et al., 2019*). However, the contributions of BAMs to CNS integrity and neurodegeneration remain to be elucidated.

To investigate the influence of AD on the macrophage CNS landscape, we analyzed the phenotype and turnover kinetics of different myeloid cells in the brain parenchyma and border-associated tissues. We used a novel AD mouse model generated by back-crossing $App^{NL-G-F}$ knock-in (APP-KI) mice (*Saito et al., 2014*) with a $Kit^{MerCreMer}/R26^{YFP}$ fate-mapping mouse strain (*Sheng et al., 2015*) to monitor bone marrow (BM)-driven replenishment of brain myeloid cells during disease progression. We also investigated the origins of CD11c[+] microglia since the ontogeny of 'activated' microglia in AD is still a matter of debate.

## Results and discussion

In this study, we conducted a comprehensive fate-mapping analysis of murine brain microglia and BAMs and their turnover kinetics during the progression of AD.

Mouse models of AD have been instrumental in clarifying the cellular and molecular mechanisms underlying this irreversible brain disorder. Transgenic mice overexpressing proteins linked to familial AD (5xFAD), single mutant amyloid precursors (APP), or double mutant APP and presenilin (APP-PS1) have been used in many studies (*Sasaguri et al., 2017*). In our study, we exploited an APP-KI mouse strain expressing a mutant form of humanized APP, the parent protein of Aβ, knocked in under the control of the endogenous promoter. This model avoids some disadvantages associated with transgenic APP models, including artifacts caused by overexpression of other APP fragments in addition to Aβ, non-physiological cell-type expression, and potential insertion site disruption (*Saito et al., 2014*). The APP-KI mouse line expresses three human AD-associated mutations that promote the progressive accumulation of Aβ through its increased production of Aβ, particularly the more toxic Aβ42 form, as well as increasing Aβ aggregation and reducing degradation. APP-KI mice develop progressive Aβ accumulation from 2 months of age, thus mimicking several aspects of human AD, including microgliosis and synaptic loss (*Figure 1—figure supplement 1A*).

First, we characterized the myeloid cell landscape in different brain regions of healthy young WT and APP-KI mice (2 months) as well as aged WT and APP-KI mice (12 months) (*Figure 1—figure supplement 2*). In our multiparameter flow cytometry and uniform manifold approximation and projection analyses, we included a panel of myeloid markers to delineate microglia, activated CD11c[+] microglia, monocyte-derived macrophages (MdCs), F4/80[int]CD11a[+] infiltrating macrophages (*Shukla et al., 2019*), border-associated macrophages (BAMs), neutrophils, monocytes and, eosinophils.

P2RY12[+] microglia were the main CD45[int] F4/80[hi] cell population in the brain parenchyma of all mice and their absolute numbers are significantly enhanced in brains of 12-month-old APP-KI mice due to microgliosis occurring during the development of AD (*Sevenich, 2018*). AD mice showed also a substantial increase in the absolute numbers of P2RY12[low]CD11c[+] activated microglia, which were absent in the brain of young mice and constituted only a minor fraction in aged mice (*Figure 1A–C and E*). Immunofluorescent analysis showed that CD11c[+] activated microglia aggregate around Aβ amyloid plaques (*Figure 1D*) In accordance with reports of the appearance of activated microglia during neurodegeneration in other AD transgenic mouse models, such as 5xFAD and APP/PS1

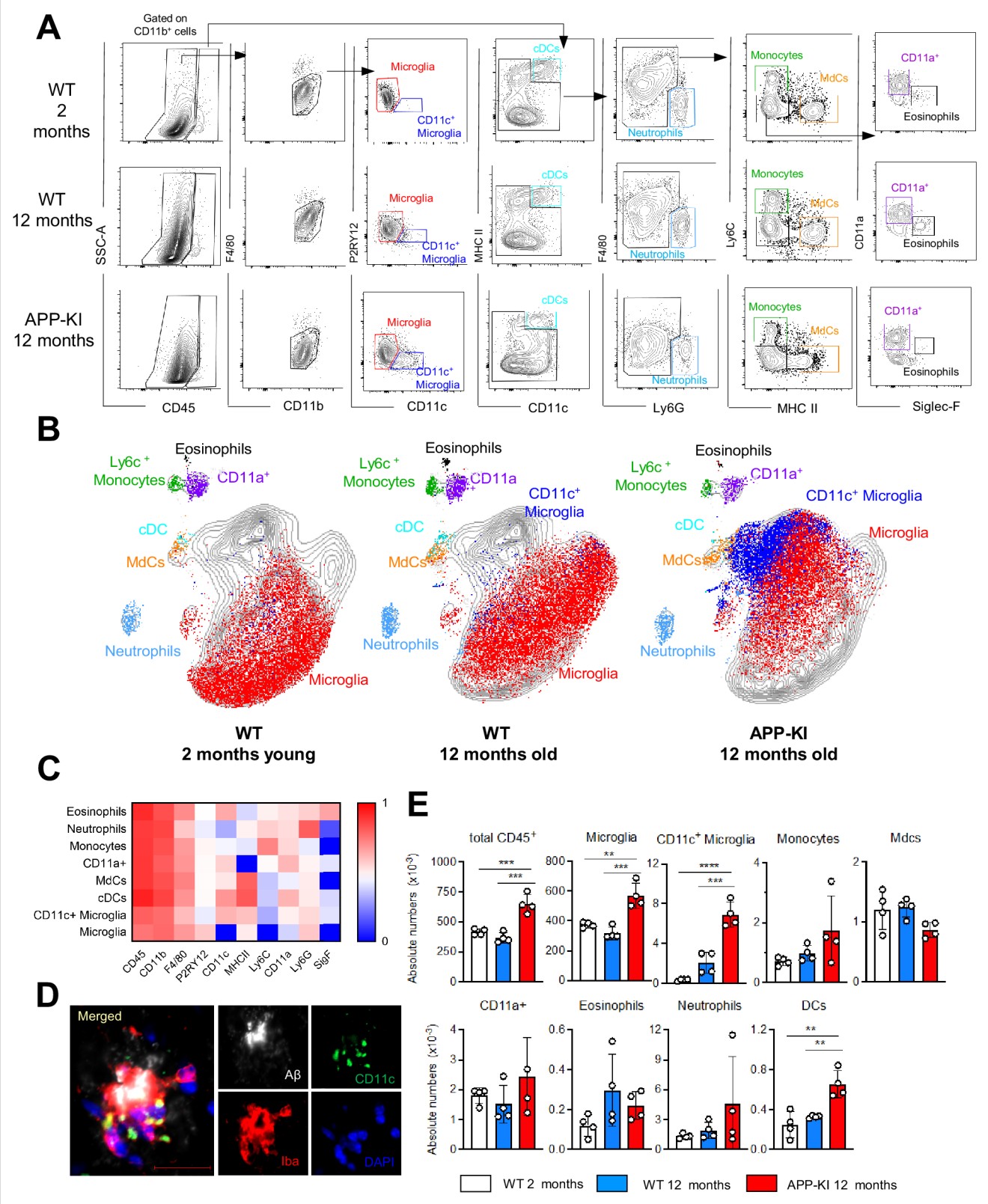

**Figure 1.** Myeloid cell profiling in healthy and AD parenchymal brains. (**A**) Representative gating strategy used to visualize distinct myeloid cell subpopulations in young WT (aged 2 months), WT (aged 12 months), and diseased APP-KI (aged 12 months) mice. Microglia: CD45$^{int}$F4/80$^{hi}$P2RY12$^{+}$CD11c$^{-}$; activated microglia: CD45$^{int}$F4/80$^{hi}$P2RY12$^{low}$CD11c$^{+}$; cDCs: CD45$^{hi}$CD11c$^{hi}$MHCII$^{+}$; neutrophils: CD45$^{hi}$CD11b$^{hi}$Ly6G$^{+}$; monocytes: CD45$^{hi}$F4/80$^{int}$CD11b$^{int}$Ly6C$^{+}$MHCII$^{-}$; monocyte-derived macrophages (MdCs): CD45$^{hi}$F4/80$^{int}$CD11b$^{int}$Ly6C$^{-}$MHCII$^{+}$; CD11a$^{+}$ cells:

*Figure 1 continued on next page*

*Figure 1 continued*

CD45$^{hi}$F4/80$^{int}$CD11b$^{int}$Ly6C$^-$MHCII$^-$CD11a$^+$ and eosinophils: CD45$^{hi}$F4/80$^{int}$CD11b$^{int}$Siglec-F$^+$. (**B**) UMAP analysis displaying 30,000 randomly sampled cells from young WT, aged WT, and APP-KI mouse brains analyzed by multicolor flow cytometry (n=3 mice/group). (**C**) Heatmap demonstrating the mean fluorescent intensity of 10 myeloid lineage markers across eight different brain parenchyma myeloid cell populations. The color in the heatmap varies from blue for lower expression to red for higher expression. (**D**) Representative confocal image of Iba-1$^+$ (red), CD11c$^+$ (green) microglia accumulating around Aβ plaques (white) in a 12-month-old APP-KI mouse brain. Blue visualizes DAPI positive nuclei. Scale bar, 20 μm. (**E**) Bar charts with individual dots illustrating the absolute numbers of different myeloid cell populations within the total parenchymal brain CD45$^+$ cell population. Each dot represents the percentage of cells obtained from one brain (n=4 mice/group). Young mice: gray, aged mice: blue, and APP-KI mice: red. Samples were analyzed by two-way ANOVA. **p<0.01; ***p<0.001; ****p<0.0001. For clarity, non-significant values are not shown. AD, Alzheimer's disease.

The online version of this article includes the following source data and figure supplement(s) for figure 1:

**Source data 1.** Absolute numbers of different myeloid cell populations within the total parenchymal brain CD45$^+$ cell population.

**Figure supplement 1.** APP-KI AD mouse model.

**Figure supplement 1—source data 1.** Frequency of activated CD11c$^+$ microglia obtained from WT and AAP-KI mice aged 3, 6 and 9 months.

**Figure supplement 2.** No difference in myeloid cells between 8-week young WT and APP-KI mice.

**Figure supplement 2—source data 1.** Absolute numbers of different myeloid cell populations within the total parenchymal brain CD45$^+$ cell population of young WT and APP-KI (2 months).

(*Keren-Shaul et al., 2017*; *Mrdjen et al., 2018*), this phenotype was recapitulated in APP-KI mice in our study (*Figure 1—figure supplement 1B,C*). The remaining myeloid cell subsets detected in the brain parenchyma, such as neutrophils (Ly6G$^+$), monocytes (Ly6C$^{hi}$), F4/80$^{int}$ Ly6C$^{neg}$MHCII$^{hi}$ MdCs, peripheral-derived myeloid F4/80$^{int}$MHCII$^-$CD11a$^+$ cells (*Shukla et al., 2019*), eosinophils (Siglec-F$^+$), and cDCs (CD11c$^{hi}$MHCII$^{hi}$), were mainly restricted to the CD45$^{hi}$ gate (*Figure 1A*). Surprisingly, we did not observe an enhanced infiltration of BM-derived inflammatory cells in the brain during disease progression. Even in aged APP-KI mice, the infiltration by Ly6C$^{hi}$ monocytes and Ly6G$^+$ neutrophils was comparable to that in age-matched controls (*Figure 1A–B and E*).

In all three separately analyzed CNS border-associated tissues, a predominant F4/80$^{hi}$CD206$^+$ BAM fraction could be further separated into MHCII$^+$ and MHCII$^{low}$ subpopulations, which is consistent with previous reports (*Van Hove et al., 2019*; *Figure 2A–D*, *Figure 2—figure supplement 1*). A large population of CD206$^+$MHCII$^{low}$ BAMs was preferentially localized in the SDM and CP, whereas these cells were less abundant in the DM. Interestingly, a significant transition from CD206$^+$MHCII$^{low}$ to CD206$^+$MHCII$^+$ BAMs was observed with aging and AD progression (*Figure 2B and C*). Similarly, increased MHCII$^+$ CNS-associated macrophages were also observed during EAE induced neuroinflammation (*Jordao et al., 2019*). All other myeloid cell populations, including neutrophils, monocytes, MdCs, CD11a$^+$ cells, eosinophils, and DCs, were present in each tissue, although at different frequencies (*Figure 2A–D*, *Figure 2—figure supplements 1 and 2*). Similar to the brain parenchyma, monocyte and neutrophil frequencies were not augmented in any of the three border regions of the aged WT and APP-KI mice, which excludes the possibility that an enhanced inflammatory cell infiltration is caused by aging or neurodegenerative disease progression (*Figure 2A–C*, *Figure 2—figure supplements 1 and 2*).

To investigate the origins and replenishment kinetics of distinct brain macrophage subpopulations, we crossed the APP-KI mouse line with the *Kit*$^{MerCreMer}$/*R26*$^{YFP}$ fate-mapping mouse strain. The resulting APP-KI/*Kit*$^{MerCreMer}$/*R26*$^{YFP}$ inducible adult fate-mapping mouse model allowed us to irreversibly label Kit-expressing BM-progenitors and trace them in different brain regions during the progression of AD. To induce the YFP label in Kit-expressing BM progenitor cells, APP-KI/*Kit*$^{MerCreMer}$/*R26*$^{YFP}$ mice and the corresponding *Kit*$^{MerCreMer}$/*R26*$^{YFP}$ controls were administered TAM at 2, 4, 6, and 8 months of age. Parenchymal and non-parenchymal brain cells were isolated separately and analyzed by multicolor flow cytometry.

In the parenchyma of 10-month-old mice, a strong YFP signal was limited to CD45$^{hi}$ cells and was barely detectable in the CD45$^{int}$ fraction, which included the microglia and activated CD11c$^+$ microglia, in both WT and AD mice (*Figure 3A–C*). In fact, microglia showed minimal YFP-labeling, which further confirmed their embryonic origin and BM-independence (*Sheng et al., 2015*). During AD progression, microglial YFP-labeling remained minimal, indicating that the ongoing neurodegeneration did not promote their replenishment by BM-derived cells. Similarly, activated CD11c$^+$ microglia, which were increasingly formed during AD progression (*Figure 1—figure supplement 1B-C*), maintained

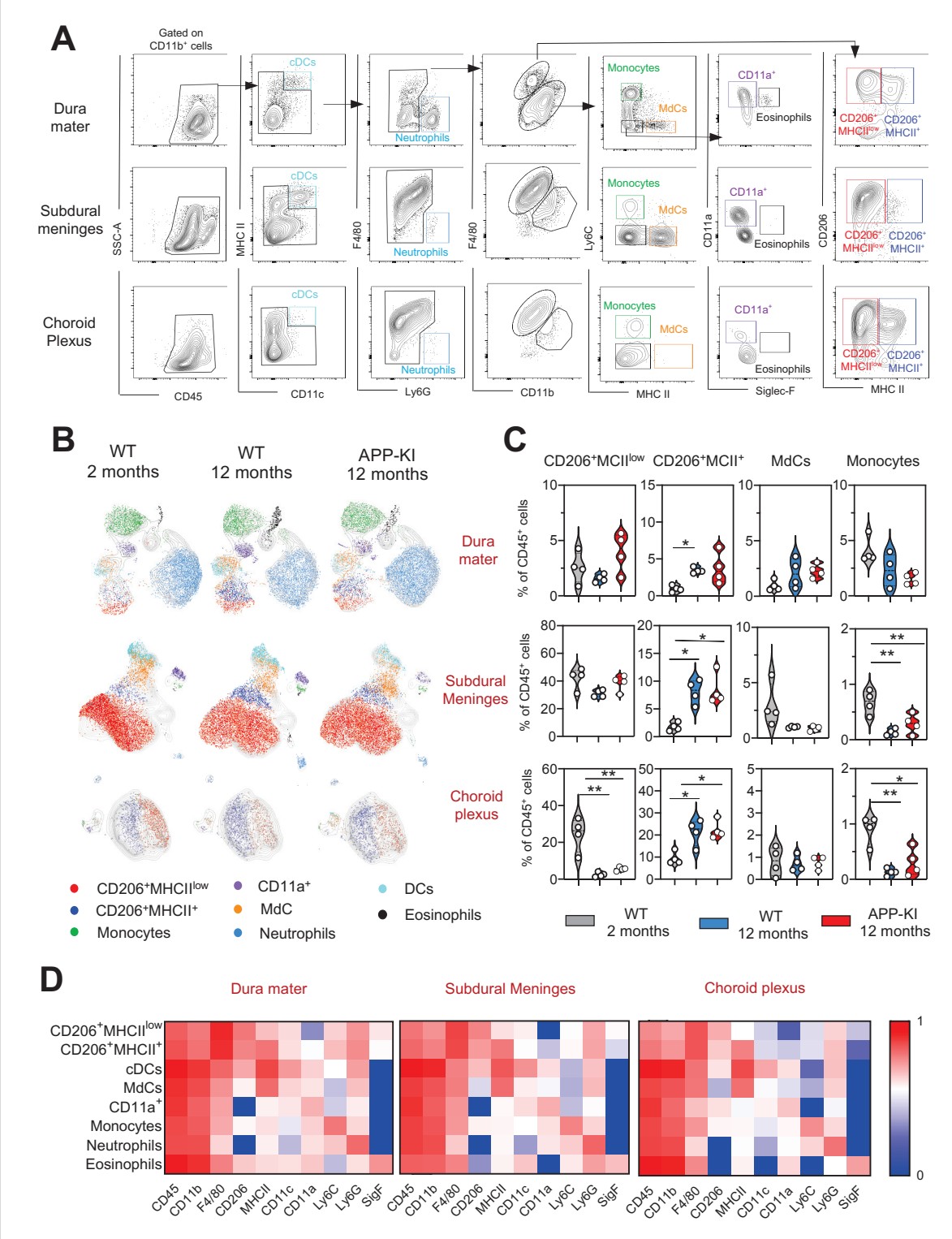

**Figure 2.** Characterization of myeloid cells in distinct brain border regions in health and AD. (**A**) Representative gating strategy used to visualize distinct myeloid cell subpopulations in the dura mater (DM), subdural meninges (SDM), and choroid plexus (CP). Young WT mice (aged 2 months) were analyzed. Cell classification as shown in the legend of **Figure 1**. (**B**) UMAP analysis displaying 8500 (DM), 13,000 (SDM), and 3700 (CP) randomly sampled cells from young WT, aged WT, and aged APP-KI DM, SDM, and CP analyzed by multicolor flow cytometry (n=9 mice). (**C**) Violin plots with individual dots illustrating the frequency of BAMs (CD206+MHCII^low and CD206+MHCII+), Mdc, and monocytes within the total non-parenchymal CD45+ cell population of the DM, SDM, and CP. Young mice: gray, aged mice: blue, and APP-KI mice: red. Each dot represents the percentage of cells obtained

*Figure 2 continued on next page*

Figure 2 continued

from pooled border regions (n=2–3 pooled mice). Samples were analyzed by two-way ANOVA. *p<0.05; **p<0.01. For clarity, non-significant values are not shown. (D) Heatmap demonstrating the mean fluorescent intensity of 10 myeloid lineage markers across eight different myeloid cell populations in DM, SDM, and CP. The colour in the heatmap varies from blue for lower expression to red for higher expression. AD, Alzheimer's disease; UMAP, uniform manifold approximation and projection.

The online version of this article includes the following source data and figure supplement(s) for figure 2:

**Source data 1.** Frequency of BAMs (CD206$^+$MHCII$^{low}$ and CD206$^+$MHCII$^+$), Mdc, and monocytes within the total non-parenchymal CD45$^+$ cell population of the DM, SDM, and CP.

**Figure supplement 1.** Characterization of myeloid cells in distinct brain border regions in health and AD.

**Figure supplement 2.** Violin plots with individual dots illustrating the frequency of different myeloid cell populations (CD11a$^+$ macrophages, DCs, neutrophils, and eosinophils) within the total non-parenchymal CD45$^+$ of the DM, SDM, and CP.

**Figure supplement 2—source data 1.** Frequency of different myeloid cell populations (CD11a$^+$ macrophages, DCs, neutrophils, and eosinophils) within the total non-parenchymal CD45$^+$ of the DM, SDM, and CP.

a low YFP-labeling profile, suggesting that despite their phenotypic shift, these cells preserved their embryonic signature and were not replaced by BM-derived cells during evolution of the disease (*Figure 3A–C*). Differently, the kinetics of monocyte replacement was extremely rapid, with all monocytes YFP-labeled 2 months after induction, indicating that these cells are exclusively BM-derived (*Figure 3A and B*, lower panel, middle). In contrast, fetal-derived MdCs were gradually replaced by BM-derived cells over time and were fully replenished at 8 months post-induction, with no significant difference between WT and AD mice (*Figure 3A and B*, lower panel, left). Likewise, CD11a$^+$ macrophages were clearly BM-derived and showed high YFP-labeling and turnover kinetics comparable to monocytes and neutrophils (*Figure 3A and B*, lower panel, right).

To verify whether activated CD11c$^+$ microglia retained their fetal lineage, we performed E7.5 embryonic labeling of APP-KI/*Kit*$^{MerCreMer}$/*R26*$^{YFP}$ mice and analyzed microglia and activated CD11c$^+$ microglia obtained from disease-affected offspring aged 9 months. Despite the AD-associated pathology, we not only confirmed that activated CD11c$^+$ microglia maintained their embryonic origin, but showed that microglia and activated CD11c$^+$ microglia share the same yolk sac origin, whereas all other tested myeloid cells (monocytes, Mdc, CD11a$^+$ macrophages, and neutrophils) did not arise from embryonic progenitors of the yolk sac (*Figure 3D*).

Using our fate-mapping mouse model, we then analyzed myeloid cells residing in distinct CNS border-associated regions. In adult fate-mapping, non-parenchymal monocytes, MdCs, and CD11a$^+$ macrophages were swiftly replaced by BM cells, with 100% YFP-positivity at 2 months post-induction (*Figure 4*, right, *Figure 4—figure supplement 1*). In contrast, both CD206$^+$ BAM subpopulations showed weaker YFP-labeling compared to other macrophages such as MdCs, which were already fully replenished within 2 months in all three regions (*Figure 4*). In the DM, YFP-labeling was significantly higher in CD206$^+$MHCII$^+$ BAMs than that in CD206$^+$MHCII$^{low}$ BAMs (*Figure 4*, upper panel). Dural CD206$^+$MHCII$^+$ BAMs were characterized by a slow, but consistent BM-cell replacement over time, reaching approximately 35% YFP-positivity by 8 months post-induction. In contrast, the turnover rate of both BAM populations located in the SDM was markedly slower (*Figure 4*, middle panel), which suggests that their niche is less accessible to BM-dependent replenishment. With its fenestrated blood vessels, the DM has a greater capacity to support peripheral cell traffic. In addition, skull BM-derived monocytes, which seed DM through tiny vascular connecting corridors (*Cugurra et al., 2021*), can also contribute to the refilling of dural BAMs. In contrast, the strong tight junctions of the blood vessels in the SDM limit cell exchange with the periphery (*Mastorakos and McGavern, 2019*). Previous studies have shown a rapid BAM turnover in the CP supported by partial input from the circulation (*Goldmann et al., 2016*; *Van Hove et al., 2019*). However, although both MHCII$^{low}$ and MHCII$^+$CD206$^+$ BAMs displayed a dual origin in our fate-mapping mouse model, their replenishment from BM-derived cells was extremely slow, with the YFP signal reaching only 10–15% by 8 months after induction. More than 85% of CP BAMs retained their embryonic phenotype (*Figure 4*, lower panel). Consistent with the lowest levels of YFP labeling observed in MHCII$^{low}$CD206$^+$ BAMs located in SDM and CP (*Figure 4*, left panels) we found a significantly increased YFP labeling only in this particular BAM subpopulation obtained from the progeny of E7.5 TAM-treated mice (SDM MHCII$^{low}$CD206$^+$

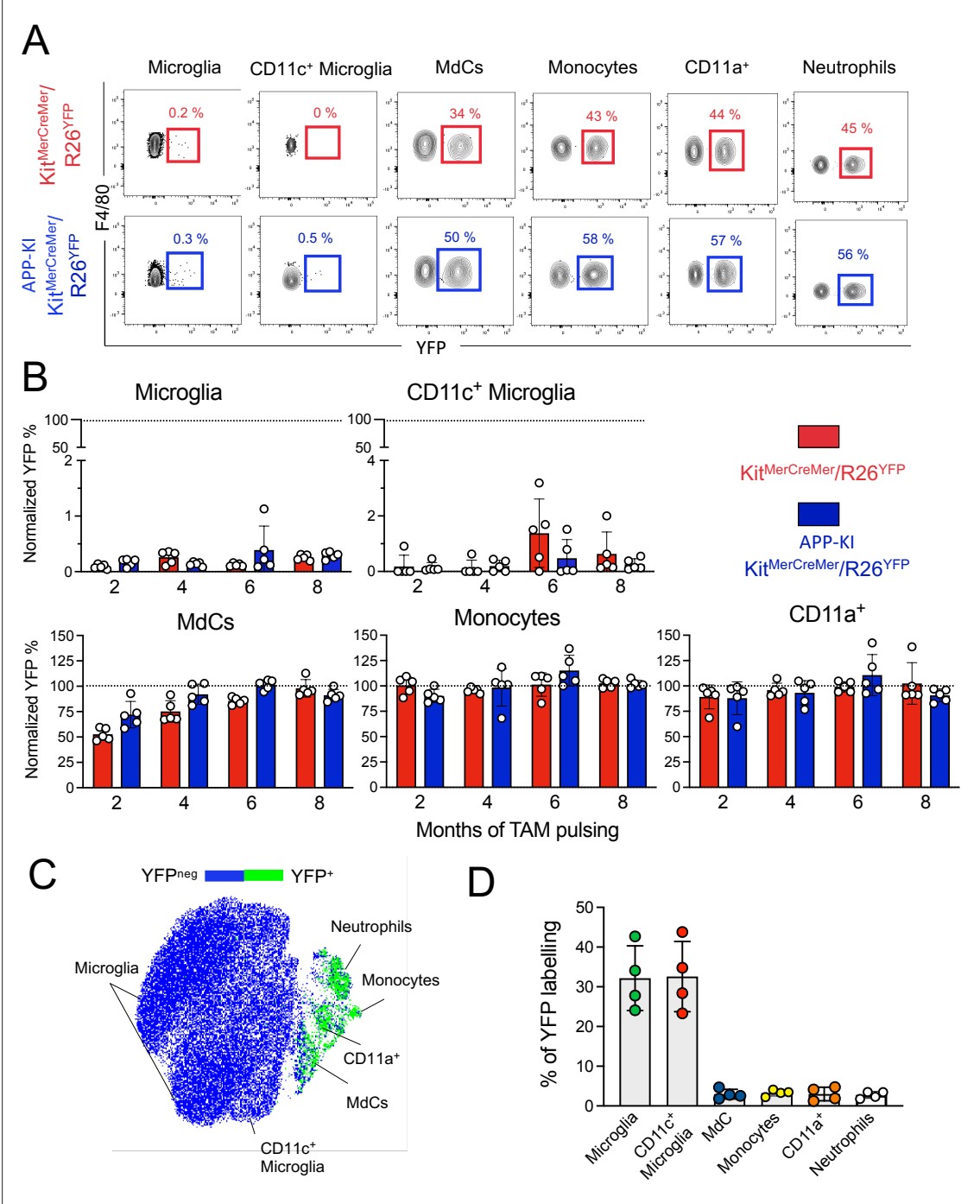

**Figure 3.** Microglia and disease-associated microglia maintain their embryonic origin during Alzheimer's disease. *Kit*^MerCreMer^/*R26*^YFP^ and APP-KI/ *Kit*^MerCreMer^/*R26*^YFP^ mice (aged 2, 4, 6, and 8 months) received 4 mg tamoxifen (TAM) by oral gavage for 5 consecutive days and were sacrificed at 10 months of age. (**A**) Representative flow cytometry contour plots indicating the YFP-labeling of parenchymal microglia, activated CD11c+ microglia, MdCs, CD11a+ cells, monocytes, and neutrophils obtained from *Kit*^MerCreMer^/*R26*^YFP^ and APP-KI/*Kit*^MerCreMer^/*R26*^YFP^ mice ( 8-month labeled mice). (**B**) Bar charts with individual data points showing the percentage of YFP+ parenchymal microglia, activated CD11c+ microglia, MdCs, monocytes, and CD11a+ cells obtained from *Kit*^MerCreMer^/*R26*^YFP^ (red bars) and APP-KI/*Kit*^MerCreMer^/*R26*^YFP^ mice (blue bars) after normalization to the percentage of YFP+ neutrophils. Data represent the mean± SD (n=5 mice). Student's t-test (two-tailed). For clarity, non-significant values are not shown. (**C**) UMAP representation showing the YFP-labeled myeloid cell populations in green and the YFP-negative fraction in blue. (**D**) Embryonic fate-mapping. A single pulse of tamoxifen was administered to APP-KI/*Kit*^MercreMer^/*R26*^YFP^ pregnant mice at E7.5. Offspring were analyzed at 9 months of age. Bar charts with individual data show the percentages of YFP-labeled parenchymal microglia, activated CD11c+ microglia, MdCs, monocytes, CD11a+ cells and neutrophils. Data represent the mean± SD (n=4 mice). UMAP, uniform manifold approximation and projection.

*Figure 3 continued on next page*

*Figure 3 continued*

The online version of this article includes the following source data for figure 3:

**Source data 1.** Frequency of YFP+ parenchymal microglia, activated CD11c+ microglia, MdCs, monocytes, and CD11a+ cells.

BAMs: 10% ; CP MHCII^low^CD206+ BAMs: 15%) (*Figure 4—figure supplement 2*). These data indicate that some of these SDM and CP MHCII^low^CD206+ BAMs are descendants of yolk sac precursors.

The lack of a significant difference in the YFP-labeling profiles of BAMs from healthy or AD mice indicates that the development of AD neither supports nor accelerates BAM replacement by BM-progenitors (*Figure 4*).

The contribution of monocytes to the pool of microglia is still unclear. The monocyte-to-microglia transition occurs in the brain, but only under certain conditions of inflammation or injury, such as meningitis (*Djukic et al., 2006*) and neonatal stroke (*Chen et al., 2020*). However, the original yolk sac embryonic microglia identity is preserved during experimental autoimmune encephalomyelitis

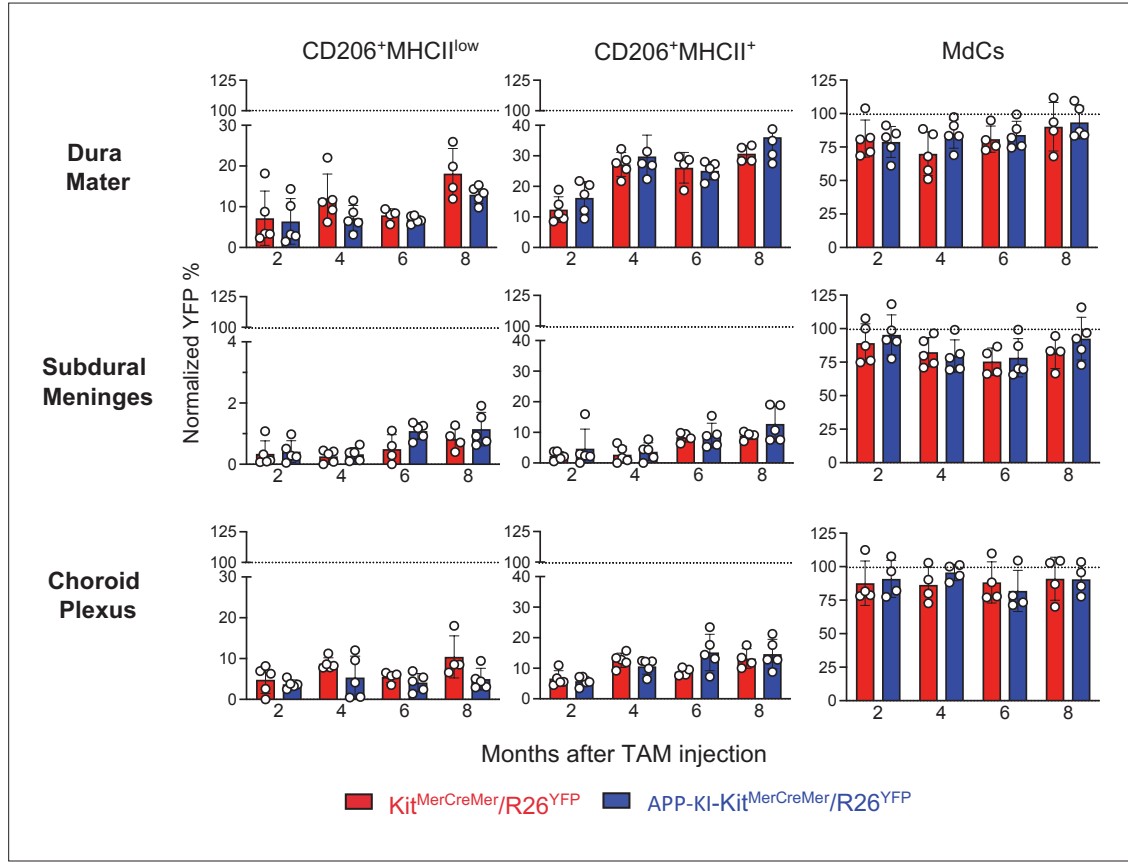

**Figure 4.** AD does not accelerate the turnover kinetics of border-associated macrophages (BAMs). Mice were treated and analyzed as described in the legend of *Figure 3*. Bar charts with individual data points showing the percentages of YFP+ non-parenchymal BAMs (CD206+MHCII^low^ and CD206+MHCII+ cells) and MdCs obtained from *Kit*^MerCreMer^/*R26*^YFP^ (red bars) and APP-KI/*Kit*^MerCreMer^/*R26*^YFP^ mice (blue bars) after normalization to the percentage of YFP+ neutrophils. Upper panel: DM; middle panel: SDM; and lower panel: CP. Data represent the mean± SD (n=4 samples of 2–3 pooled mice). Student's t-test (two-tailed). For clarity, non-significant values are not shown. AD, Alzheimer's disease; CP, choroid plexus; DM, dura mater; MdC, monocyte-derived macrophage; SDM, subdural meninges.

The online version of this article includes the following source data and figure supplement(s) for figure 4:

**Source data 1.** Percentages of YFP+ non-parenchymal BAMs (CD206+MHCII^low^ and CD206+MHCII+ cells) and MdCs.

**Figure supplement 1.** Bar charts with individual data points showing the percentages of YFP+ non-parenchymal CD11a+ cells and monocytes obtained from *Kit*^MerCreMer^/*R26*^YFP^ (red bars) and APP-KI/*Kit*^MerCreMer^/*R26*^YFP^ mice (blue bars) after normalization to the percentage of YFP+ neutrophils.

**Figure supplement 1—source data 1.** Percentages of YFP+ non-parenchymal CD11a+ cells and monocytes.

**Figure supplement 2.** Embryonic E7.5 fate-mapping of border-associated myeloid cells.

**Figure supplement 2—source data 1.** Embryonic E7.5 fate-mapping of border-associated myeloid cells.

(*Ajami et al., 2011*; *Jordao et al., 2019*), although infiltrating monocytes are recruited to the brain during the progression of this neuroinflammatory disease. In an elegant parabiosis experiment using two different mouse models of AD (APPPS1-21 and 5xFAD), it was shown that peripheral monocytes do not contribute to the microglial pool surrounding the Aβplaques (*Wang et al., 2016*). Here, we extended our analysis and demonstrated that, in the steady state, microglia, activated CD11c⁺ microglia and BAMs are a remarkably stable embryonic-derived population and their turnover remains unaffected in AD; the progression of this neurodegenerative disease does not promote or sustain their replacement by BM-derived progenitors. Furthermore, we confirmed that the transformation from homeostatic microglia to activated CD11c⁺ microglia is not mediated by infiltrating inflammatory BM-marrow derived cells, but imprinted by the local brain microenvironment that is severely perturbed by the AD pathology. This new understanding will be valuable in manipulating the transition of microglia to activated microglia as a therapeutic strategy in AD.

# Materials and methods

## Key resources table

| Reagent type (species) or resource | Designation | Source or reference | Identifiers | Additional information |
|---|---|---|---|---|
| Antibody | Anti-CD45 (rat, monoclonal) | BD Biosciences | Cat#: 748719; RRID:AB_2873123 | FACS (1:600; 100 µl per test) |
| Antibody | Anti-Siglec-F (rat, monoclonal) | BD Biosciences | Cat#: 746668; RRID:AB_2743940 | FACS (1:600; 100 µl per test) |
| Antibody | Anti-CD11b (rat, monoclonal) | BioLegend | Cat#: 101226; RRID:AB_830642 | FACS (1:600; 100 µl per test) |
| Antibody | Anti-CD11b (rat, monoclonal) | BD Biosciences | Cat#: 565976; RRID:AB_2738276 | FACS (1:600; 100 µl per test) |
| Antibody | Anti-F4/80 (rat, monoclonal) | BioLegend | Cat#: 123110; RRID:AB_893486 | FACS (1:600; 100 µl per test) |
| Antibody | Anti-F4/80 (rat, monoclonal) | BioLegend | Cat#: 123114; RRID:AB_893490 | FACS (1:600; 100 µl per test) |
| Antibody | Anti-Ly6c (rat, monoclonal) | BioLegend | Cat#: 128030; RRID:AB_2562617 | FACS (1:600; 100 µl per test) |
| Antibody | Anti-I-A/I-E (rat, monoclonal) | BioLegend | Cat#: 107636; RRID:AB_2734168 | FACS (1:1000; 100 µl per test) |
| Antibody | Anti-CD11c (hamster, monoclonal) | BioLegend | Cat#: 117318; RRID:AB_493568 | FACS (1:600; 100 µl per test) |
| Antibody | Anti-CD11c (hamster, monoclonal) | BioLegend | Cat#: 117336; RRID:AB_2565268 | FACS (1:600; 100 µl per test) |
| Antibody | Anti-P2RY12 (rat, monoclonal) | BioLegend | Cat#: 848006; RRID:AB_2721469 | FACS (1:600; 100 µl per test) |
| Antibody | Anti-CD11a (rat, monoclonal) | Invitrogen | Cat#: 48-0111-82; RRID:AB_11064445 | FACS (1:600; 100 µl per test) |
| Antibody | Anti-CD206 (rat, monoclonal) | BioLegend | Cat#: 141734; RRID:AB_2629637 | FACS (1:600; 100 µl per test) |
| Antibody | Anti-Ly6G (rat, monoclonal) | BioLegend | Cat#: 127606; RRID:AB_1236494 | FACS (1:600; 100 µl per test) |
| Antibody | Anti-Iba-1 (rabbit, polyclonal) | Fujifilm Wako Shibayagi | Cat#: 019-19741; RRID:AB_839504 | IHC (1:200; 100 µl per test) |
| Antibody | Anti-CD11c (hamster, monoclonal) | Self-made | N/A | IHC (1:100; 100 µl per test) |
| Antibody | Anti-amyloid beta (mouse, monoclonal) | IBL-Immuno-Biological-Laboratories Co. | Cat#: 10323; RRID:AB_10707424 | IHC (1:200; 100 µl per test) |
| Antibody | Anti-IgG (goat, polyclonal) | BioLegend | Cat#: 405502 RRID:AB_315020 | IHC (1:200; 100 µl per test) |

*Continued on next page*

*Continued*

| Reagent type (species) or resource | Designation | Source or reference | Identifiers | Additional information |
|---|---|---|---|---|
| Antibody | Anti-IgG (donkey, polyclonal) | BioLegend | Cat#: 406418 RRID:AB_2563306 | IHC (1:200; 100 µl per test) |
| Antibody | Anti-IgG (goat, polyclonal) | BioLegend | Cat#: 405308 RRID:AB_315011 | IHC (1:200; 100 µl per test) |
| Antibody | Anti CD16/32 (rat, monoclonal) | Self-made | N/A | Blocking step (1:100; 1000 µl per sample) |
| Chemical compound, drug | DAPI | Thermo Fisher Scientific | Cat#: D1306 | (1:1000) |
| Chemical compound, drug | Collagenase D | Roche | Cat#: 11088882001 | 1 mg/ml |
| Chemical compound, drug | Dispase II | Gibco | Cat#: 17105041 | 2 U/ml |
| Chemical compound, drug | DNase | Roche | Cat#: 04536282001 | 2 U/ml |
| Chemical Compound, drug | Percoll | Merck | Cat#: P4937-500ML | |
| Chemical compound, drug | Progesterone | Sigma-Aldrich | Cat#: P0130 | 1 mg/mouse |
| Chemical compound, drug | Tamoxifen | Sigma-Aldrich | Cat#: T5648 | For adult labeling, 4 mg TAM for 5 consecutive days by oral gavage; For embryo labeling, pregnant mice (E7.5) were injected via oral gavage once with 2 mg TAM |
| Chemical compound, drug | IMDM | Thermo Fisher Scientific | Cat#: 12440046 | |
| Software, algorithm | FlowJo | TreeStar | FlowJo 10.6 RRID:SCR_008520 | |
| Software, algorithm | GraphPad Prism | GraphPad Software | GraphPad 9.0 RRID:SCR_002798 | |
| Strain, strain background (*mouse*) | APP$^{NL-G-F}$ (called APP-KI) | Japan *Saito et al., 2014* | | |
| Strain, strain background (*mouse*) | *Kit*$^{MerCreMer}$/Rosa26-LSL-eYFP (called *Kit*$^{MerCreMer}$/*R26*$^{YFP}$) | Nanyang Technological University, Singapore *Sheng et al., 2015* | | |
| Strain, strain background (*mouse*) | APP-KI/ *Kit*$^{MerCreMer}$/*R26*$^{YFP}$ | Nanyang Technological University, Singapore | Described here | |

## Mice

Fate-mapping *Kit*$^{MerCreMer}$/*R26*$^{YFP}$ mice were generated as previously described (*Sheng et al., 2015*). The *Kit*$^{MerCreMer}$/*R26*$^{YFP}$ fate-mapping mouse line was crossed with the AD mouse model (APP$^{NL-G-F}$), a knock-in mouse line that co-expresses the Swedish (KM670/671 NL), Beyreuther/Iberian (I716F), and Arctic (E693G) mutations and mimics AD-associated pathologies, including amyloid plaques, synaptic loss, and microgliosis as well as astrocytosis (*Saito et al., 2014*); the generated mouse line was designated APP-KI/*Kit*$^{MerCreMer}$/*R26*$^{YFP}$. Mice were bred and maintained in a specific pathogen-free animal facility at the Nanyang Technological University (Singapore). All animal studies were carried out according to the recommendations of the National Advisory Committee for Laboratory Animal Research and ARF SBS/NIE 18081, and 19093 protocols were approved by the Institutional Animal Care and Use Committee of the Nanyang Technological University.

## Tamoxifen-inducible embryonic and adult fate-mapping mouse model

*Kit*$^{MerCreMer}$/*R26*$^{YFP}$ and APP-KI/*Kit*$^{MerCreMer}$/*R26*$^{YFP}$ fate-mapping mice were used to determine the turnover rates of distinct brain macrophages under normal steady-state or diseased conditions. For adult labeling, a total of 4 mg tamoxifen (TAM) (Sigma-Aldrich, St. Louis, MO) per mouse was administered for 5 consecutive days by oral gavage as previously described (*Sheng et al., 2015*). Mice were sacrificed at different time points for brain collection, subsequent cell isolation,

and multiparameter flow cytometric cell analysis. For embryonic labeling, $Kit^{MerCreMer}/R26^{YFP}$ and APP-KI/$Kit^{MerCreMer}/R26^{YFP}$ fate-mapping mice were mated overnight and separated early the next morning. Pregnant mice (E 7.5) received one dose of 2 mg TAM with 1 mg progesterone via oral gavage.

## Isolation of microglia and border-associated macrophages

Brains were removed and DM, SDM, and CP were carefully separated from the brain parenchyma. For the DM isolation, the dorsal part of the skull was removed and the dura was peeled away from the skull cap and placed in 2% fetal bovine serum (FBS) in Iscove's modified Dulbecco's medium (IMDM). The SDM was micro-dissected using micro suture forceps and placed on ice-cold 2% FBS IMDM. For collection of the CP, the ventricles were exposed and the CP was carefully micro-dissected from the lateral ventricles and placed on 2% FBS IMDM.

All tissues were cut into small pieces, which were subsequently incubated with digestion buffer (IMDM supplemented with 2% FBS, 1 mg/ml Collagenase D [Roche], 2 U/ml DNase I [Life Technologies], and Dispase II [Roche]). The digested brain parenchyma, DM, SDM, and CP were separately homogenized with a syringe and the resulting homogenous cell suspensions were filtered through a 40 μm cell strainer. Only the parenchyma cells were resuspended in a 40% Percoll (GE Healthcare Life Sciences) and centrifuged at 700×$g$ for 10 min. All obtained cell pellets were resuspended in 0.89% $NH_4CL$ lysis buffer for 5 min at room temperature (RT) to remove contaminating red blood cells. After centrifugation at 350×$g$ for 5 min, the supernatant was discarded and cell pellets were collected for flow cytometry staining.

## Flow cytometry staining

Isolated cells were pre-incubated with 10 μg/ml anti-Fc receptor antibody (2.4G2) on ice for 20 min. Subsequently, cells were stained with different antibodies for 20 min on ice. After washing, cells were further stained with DAPI to exclude dead cells. Finally, cells were washed and resuspended in PBS/ 2% FBS for analysis on a five-laser flow cytometer (FACSymphony A3 Cell Analyser, BD Biosciences, San Jose, CA). Data were analyzed with FlowJo software (TreeStar, Ashland, OR).

## Immunofluorescence staining and microscopy

Mice were perfused with 10 ml 1× PBS and 20 ml 4% paraformaldehyde (PFA). After extraction, the brains were post-fixed with 4% PFA for 24 hr. Tissues were then transferred to 15% sucrose solution for 24 hr, followed by immersion in 30% sucrose solution for another 24 hr. After that, the brains were then embedded in the Optimal cutting temperature compound and cut into 5 μm thick sections. For the immunofluorescence staining, sections were dried at 37°C for 10 min, then wash with 1× PBS at RT for 5 min. Sections were incubated with 5% BSA at RT for 30 min, followed by primary antibody staining at 4°C for overnight (mouse anti-82E1 [1:200], hamster anti-CD11c [1:100], and rabbit anti-Iba-1 [1:200]), three times washings with 1× PBS and an incubation with the correspondent secondary antibodies (Goad anti-hamster FITC [1:200], Donkey anti-rabbit Alexa Fluor 594 [1:200], and Goat anti-mouse APC [1:200]). Stained sections were then incubated with DAPI for 10 min and mounted with fluorescence mounting medium and visualized by fluorescent microscopy (Zeiss). Images were post-processed by Zen software.

## Statistical analysis

Statistical analysis was performed using GraphPad Prism 9.0.1 software (GraphPad Software, La Jolla, CA). All values were expressed as the mean± standard deviation as indicated in the figure legends. Samples were analyzed by Student's t-test (two-tailed) or two-way ANOVA. A p-value <0.05 was considered to indicate statistical significance. The number of animals used per group is indicated in the figure legends as 'n'.

## Acknowledgements

The authors would like to thank Insight Editing London for proofreading the manuscript before submission. This work was supported by the Ministry of Education Tier 1 grant awarded to CR.

## Additional information

### Funding

| Funder | Grant reference number | Author |
|---|---|---|
| Ministry of Education - Singapore | MOE AcRF Tier 1 | Christiane Ruedl |

The funders had no role in study design, data collection and interpretation, or the decision to submit the work for publication.

### Author contributions

Xiaoting Wu, Investigation, Methodology, Visualization; Takashi Saito, Takaomi C Saido, Resources; Anna M Barron, Investigation, Methodology, Writing - review and editing; Christiane Ruedl, Conceptualization, Formal analysis, Funding acquisition, Project administration, Supervision, Validation, Visualization, Writing - original draft, Writing - review and editing

### Author ORCIDs

Xiaoting Wu http://orcid.org/0000-0002-0281-8717
Christiane Ruedl http://orcid.org/0000-0002-5599-6541

### Ethics

Mice were bred and maintained in a specific pathogen-free animal facility at the Nanyang Technological University (Singapore). All animal studies were carried out according to the recommendations of the National Advisory Committee for Laboratory Animal Research and ARF SBS/NIE 18081, and 19093 protocols were approved by the Institutional Animal Care and Use Committee of the Nanyang Technological University.

### Decision letter and Author response

Decision letter https://doi.org/10.7554/eLife.71879.sa1
Author response https://doi.org/10.7554/eLife.71879.sa2

## Additional files

### Supplementary files

• Transparent reporting form

### Data availability

All data generated or analysed during this study are included in the manuscript and supporting files. Source data files have been provided for Figures 1, 2, 3 and 4 and figure supplements.

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
