## [Decision Letter]

**Acceptance summary:**

Wu et al., describe for the first time the fate and lineage of brain myeloid cells in a model of amyloid Alzheimer's disease. In addition, the study demonstrates the low turnover of "border associated macrophages" in the brain under steady-state and pathology. This study will be of interest to immunologists, neurologists and clinicians.

**Decision letter after peer review:**

Thank you for submitting your article "Microglia and border-associated mouse macrophages maintain their embryonic origin during Alzheimer's disease" for consideration by *eLife*. Your article has been reviewed by 3 peer reviewers, and the evaluation has been overseen by a Reviewing Editor and Carla Rothlin as the Senior Editor. The reviewers have opted to remain anonymous.

Essential revisions

1. One limitation of this study is the exclusive reliance on flow cytometry and the absence of histological data. How do the authors know which cells are plaque-associated? Could it be that this is a minority of cells that might escape the bulk analysis because they escape isolation? Please provide complementary histological analysis to exclude this option. Furthermore, do P2RY12lo CD11c+ colocalize with plaques?

2. The term monocyte-derived cells (MdC) is defined in the text as F4/80int MHC2+ or F4/80int CD11a+ cells. These markers are unusual for the analysis of CNS macrophages. What are these cells? perivascular cells? or are they located within the parenchyma? The inclusion of the more general CD45 / CD11b profiling would help. Here, histological analysis is warranted, both in steady-state and during pathology, with a focus on plaques.

3. Nomenclature: The author's term P2RY12lo CD11c+ cells DAM, standing for 'disease-associated microglia', and refer to an earlier study. In their scRNAseq-based study, Keren-Shaul and colleagues introduced the term based on a comprehensive gene signature, not a two marker-based phenotype. However, this definition has been challenged, since the expression of many of the genes is observed in any activated microglia, irrespective of disease context. The community hence recommends abstaining from the too generic DAM term. The problem is highlighted on page 7, line 224, 'in the steady-state, microglia, DAMs and BAMs are a remarkably stable population'. How can disease-associated microglia exist in a steady state? To avoid confusion please use the term 'activated microglia' or best give the P2RY12lo CD11c+ phenotype. CD11c expression and P2RY12 loss is also a feature of ageing microglia, as can f.i. be seen in the 1yr old mice analysed in Figure 1A.

4. The authors could discuss the recently published Science paper (Cugurra et al., 2021) about the replenishment of meningeal monocytes from skull BM. These authors suggested that the meningeal Ly6C+ monocytes are supplied by skull bone marrow and experience a comparable high turnover rate to blood monocytes. This seems contradictory from what is proposed here: most BAMs retained the embryonic phenotype and mostly were not replaced by monocytes derived from BM including skull BM. If true, skull BM-derived Ly6C+ monocytes may in fact enter into the circulation instead of refilling the pool of BAMs. One wonders if these BAMs are ki67+ or whether they have a life-long survival.

Figures 1 and 2

5. Please include APP-KI samples at 2 months for comparison although the myeloid cell profiles in APP-KI mice would presumably retain patterns similar to WT at 2 months.

6. What is the population with CD45high/Ly6Glow/Ly6Clow/MHC IIlow/CD11a-/Siglec-F- in figure 1A? It is also specifically enriched in APP-KI.

7. Eosinophils only increased in 12-month-old WT but not in APP-KI (Figure 1C). However, according to Mrdjen et al., 2018, eosinophils were reduced in aged WT (18 months). Do the authors have an explanation for this discrepancy?

8. Have the authors stained CD3+ cells to examine the infiltration of T cells in the CD11b- population?

9. In line 139, according to supplemental Figure 1, please mention that the total population of microglia was also increased, as previously reported microgliosis with the amyloid pathology. Moreover, compared with the quantitative data (Figure 1C), does it suggest that the homeostatic microglia remain the same during ageing and amyloid pathology?

10. In Figure. 1A-C, is this analysis performed of brains without meninges and choroid plexus or total on the brain including all the borders?

11. The authors state that P2ry12+ microglia were the main CD45intF4/80hi population in the brain parenchyma. Can they show also F4/80?

12. Figure.1B and 2B, For all the different UMAPs, a heatmap of the mean marker expression for the different populations in the UMAP should be shown.

Figures 3 and 4

13. What are the Cd11a+ cells? Can this be discussed in the text?

14. In Figure 3, it appears that these mice were not perfused according to the methods and the purpose of this experiment. If the mice were perfused, please add that information in the methods.

15. In Figure 3A, please state which mice were used in these two representative flow plots?

16. In Figure 3B, the authors need to clarify that the normalization of YFP+ cells was done with neutrophils or see comment below:

17. In Figures 3 and 4, please show the percentage labelling for each population including monocytes rather than the normalization to neutrophils.

18. In Figure 3D, have they also analysed BAMs upon embryonic labelling?

19. In Figure 4, MHC II+ BAMs seem to have a significantly higher turnover rate than MHC IIlow BAM. Do the authors have an explanation for that?

20. The authors may discuss/consider that the slow turnover kinetics of BAM may be model dependent compared to other models where AD is accelerated.

Other points that need to be addressed

21. In line 153, the authors might want to cite Jordão et al., 2019, who characterized a similar increase of MHC II+ CNS-associated macrophages in an EAE model with neuroinflammation.

22. In line 172, it would be better to mention that these mice were harvested at 10 months of age.

23. The MerCreMer labelling is not 100%, but around 80%. Is this a technical constraint that the authors should bring it up in their discussion as a caveat?

---

## [Author Response]

Essential revisions1. One limitation of this study is the exclusive reliance on flow cytometry and the absence of histological data. How do the authors know which cells are plaque-associated? Could it be that this is a minority of cells that might escape the bulk analysis because they escape isolation? Please provide complementary histological analysis to exclude this option. Furthermore, do P2RY12lo CD11c+ colocalize with plaques?

We have added a new immunofluorescence staining illustrating β-amyloid plaques surrounded by Iba^+^CD11c^+^ activated microglia. Unfortunately the P2RY12 specific antibody does not work optimally in histology, hence we used the well-established microglia specific anti-Iba-1 Antibody commonly used for section staining. We have included a new panel D in Figure 1. Similar staining/result was also shown in Mrdjen et al.,; Figure S5 panel D.

2. The term monocyte-derived cells (MdC) is defined in the text as F4/80int MHC2+ or F4/80int CD11a+ cells. These markers are unusual for the analysis of CNS macrophages. What are these cells? perivascular cells? or are they located within the parenchyma? The inclusion of the more general CD45 / CD11b profiling would help. Here, histological analysis is warranted, both in steady-state and during pathology, with a focus on plaques.

We define MdCs as F4/80^int^Ly6C^neg^MHCII^+^ and apologize about the confusion in the result section which was now clarified in more detail (page 5). These cells are believed to progressively differentiate from incoming F4/80^int^Ly6C^+^MHCII^-^ monocytes through F4/80^int^Ly6C^+^MHCII^+^ intermediates (known as “monocyte waterfall”) and are described in many other organs such as intestine, adipose tissue etc.

A CD45^hi^CD11b^+^CD11a^+^ macrophage population was recently described in control and, in increased numbers, in 12-momth-old 5XFAD mice. These cells are distinct from microglia which are CD11a^neg^. CD45^hi^CD11b^+^CD11a^+^ cell are of peripheral origin and are, only on rare occasion, found near Ab plaques which are mainly surrounded Iba1^+^ microglia (Shukla et al., 2019; figure 1 c and figure 3 D). By including the CD11a marker in our antibody master mix we have identified the same cell population (both in brain parenchyma as well as border associated tissues) although we did not see any difference in their numbers when compared to aged matched control mice and APP-KI AD mice. We can confirm in our adult fate mapping analysis that these cells have a peripheral origin due to their fast turnover rate and strong YFP labelling within few weeks and they are not of yolk sac origin. We have included these additional data related to their turnover in Figure 3 B and a new supplementary Figure 4, figure supplement 1 and 2. Since perivascular cells harbour higher levels of CD45 (Goldmann et al., 2016), we can speculate that these cells, which are CD45^hi^ (Figure 1), could be perivascular located.

As requested by the reviewers we have now included the CD11b/F4/80 contour plot in Figure 1A.

We would like to re-emphasize that the main focus of this short manuscript was to define the turnover rates of tissue resident macrophages (microglia, activated microglia and BAMs) at steady state and during AD development which we have done exploiting a kit^MerCremer^ fate mapping mouse backcrossed to a APP-KI AD mouse model. Their turnover rates were measured by flow-cytometry analysis which is the state-of-the art method to do this. To make clear this point we have modified our title from “Microglia and border-associated mouse macrophages maintain their embryonic origin during Alzheimer’s disease” to “Microglia and CD206**^+^** border-associated mouse macrophages maintain their embryonic origin during Alzheimer’s disease”

As mentioned in Point 1, we have performed immunofluorescence analysis to visualize activated microglia around the β amyloid plaques. However, we feel that the request for additional histological visualization of all the other macrophage fractions by immunofluorescence goes beyond the scope of our short manuscript. We agree that this histological analysis is important but should be extensively addressed and discussed in another manuscript.

3. Nomenclature: The author's term P2RY12lo CD11c+ cells DAM, standing for 'disease-associated microglia', and refer to an earlier study. In their scRNAseq-based study, Keren-Shaul and colleagues introduced the term based on a comprehensive gene signature, not a two marker-based phenotype. However, this definition has been challenged, since the expression of many of the genes is observed in any activated microglia, irrespective of disease context. The community hence recommends abstaining from the too generic DAM term. The problem is highlighted on page 7, line 224, 'in the steady-state, microglia, DAMs and BAMs are a remarkably stable population'. How can disease-associated microglia exist in a steady state? To avoid confusion please use the term 'activated microglia' or best give the P2RY12lo CD11c+ phenotype. CD11c expression and P2RY12 loss is also a feature of ageing microglia, as can f.i. be seen in the 1yr old mice analysed in Figure 1A.

We agree with the reviewers that the DAM nomenclature can be misleading since low numbers of CD11c^+^ microglia start to appear in brains of aged mice although in very low numbers. Therefore we have, as suggested, changed the term DAM with “activated- CD11c^+^ microglia”. We emphasized that these CD11c^+^ microglia accumulate in diseased brains and were defined as DAMs in Keren-Shaul et al.

4. The authors could discuss the recently published Science paper (Cugurra et al., 2021) about the replenishment of meningeal monocytes from skull BM. These authors suggested that the meningeal Ly6C+ monocytes are supplied by skull bone marrow and experience a comparable high turnover rate to blood monocytes. This seems contradictory from what is proposed here: most BAMs retained the embryonic phenotype and mostly were not replaced by monocytes derived from BM including skull BM. If true, skull BM-derived Ly6C+ monocytes may in fact enter into the circulation instead of refilling the pool of BAMs. One wonders if these BAMs are ki67+ or whether they have a life-long survival.

We thank the reviews to mention this work which was published just when our manuscript was ready to be send out for peer-review hence was not included in the discussion.

Kipnis group observed a direct migration of monocytes and neutrophils from the skull BM into the meninges through vascular corridors independent from the blood circulation. Once skull BM derived monocytes reach the Dura, they can convert into macrophages although at very low frequency (e.g. Figure S4 G of Cugurra et al., 2021).

We do not see any discrepancy with our data. If you compare dura matter, subdural meninges and choroid plexus, dura mater BAMs display higher YFP labelling when compared to the other two regions which are more distant from the tiny connecting vascular channels. Therefore, possibly, skull-BM-derived monocytes can contribute to the refilling of dural BAMs. We have discussed this in the result/Discussion section.

Figures 1 and 25. Please include APP-KI samples at 2 months for comparison although the myeloid cell profiles in APP-KI mice would presumably retain patterns similar to WT at 2 months.

These data were included in a new Figure 1, figure supplement 2. No main differences are observed between 2-month-old WT and 2-month-old APP-KI mice. It was for us impossible to repeat the UMAP analysis for all four conditions (2-month WT, 2-month APP, 12-month WT and 12-month APP) due to lack of mice during the manuscript revision process.

6. What is the population with CD45high/Ly6Glow/Ly6Clow/MHC IIlow/CD11a-/Siglec-F- in figure 1A? It is also specifically enriched in APP-KI.

We do not know much about these cells. A scRNA seq analysis could give more insight about this population.

7. Eosinophils only increased in 12-month-old WT but not in APP-KI (Figure 1C). However, according to Mrdjen et al., 2018, eosinophils were reduced in aged WT (18 months). Do the authors have an explanation for this discrepancy?

For each myeloid subpopulation have now replaced the calculated values in % to absolute numbers. Between the three groups (young, 12-month-old WT and APP-KI mice) we do not see any significant difference in eosinophils numbers. This discrepancy between us and the work of Mrdjen et al., could possibly be explained by the age difference (12 months versus 18 months) of the tested mice. Certainly 18-month-old mice (and not 12- month-old) can be considered geriatric and lead to this observed eosinophil reduction.

8. Have the authors stained CD3+ cells to examine the infiltration of T cells in the CD11b- population?

Main focus of this manuscript are tissue resident macrophages and with minor focus on other myeloid cells therefore we would like not to include any data related to lymphocytes. Author response image 1, for perusal of the reviewers, the data obtained with respect to T cells in the brain of 2-month old WT (grey) and 12-month old WT (blue) and APP-KI mice (red).

**Author response image 1. sa2fig1:** 

9. In line 139, according to supplemental Figure 1, please mention that the total population of microglia was also increased, as previously reported microgliosis with the amyloid pathology. Moreover, compared with the quantitative data (Figure 1C), does it suggest that the homeostatic microglia remain the same during ageing and amyloid pathology?

The old Figure 1 c showing the frequency in % was misleading due to the high frequency of the main microglia fraction (> than 90%). Absolute numbers of all myeloid cells including microglia and activated CD11c^+^ microglia are now included in a new Figure 1E. Clearly APP-KI mice develop microgliosis since absolute numbers of microglia are significantly increased in APP-KI mice when compared to aged matched WT mice. We have discussed this in the result section.

10. In Figure. 1A-C, is this analysis performed of brains without meninges and choroid plexus or total on the brain including all the borders?

All analysis of brain parenchyma was performed after have separated dura mater, subdural meninges and choroid plexus.

11. The authors state that P2ry12+ microglia were the main CD45intF4/80hi population in the brain parenchyma. Can they show also F4/80?

We assume the reviewer is asking to show CD11b? We have now included this dot plot configuration (CD11b versus F4/80) in a new Figure 1A.

12. Figure 1B and 2B, for all the different UMAPs, a heatmap of the mean marker expression for the different populations in the UMAP should be shown.

As suggested by the reviewers, we have included the heat map of the mean marker expression for the different myeloid subpopulations in parenchyma and borders. Now new Figure 1C and Figure 2D*.*

Figures 3 and 413. What are the Cd11a+ cells? Can this be discussed in the text?

See answer to point 2 under essential questions.

14. In Figure 3, it appears that these mice were not perfused according to the methods and the purpose of this experiment. If the mice were perfused, please add that information in the methods.

This information was already included in Material and methods.

15. In Figure 3A, please state which mice were used in these two representative flow plots?

The relevant information was included in the legend.

16. In Figure 3B, the authors need to clarify that the normalization of YFP+ cells was done with neutrophils or see comment below:

This was clarified in the figure legend of both figures 3 and 4.

17. In Figures 3 and 4, please show the percentage labelling for each population including monocytes rather than the normalization to neutrophils.

Answer to 16 and 17: the TAM induced YFP labelling efficiency varies between experiment and mouse batches, hence a normalization is required. As published in our original paper (Sheng , Ruedl and Karjalainen Immunity, 2015) neutrophils were chosen for normalization due to their fast turnover. Here for the perusal of the reviewers in Author response image 2 (related to Figure 3) which shows the original labelling efficiency of distinct macrophage subsets, monocytes and neutrophils at different timepoints obtained from distinct WT and APP mice. (The white bars show the YFP labelling of neutrophils for each time point and mouse strain; these values are considered 100% in the performed normalization for Figure 3). For clarity reasons we still prefer to present normalized data, what was done for all our previously published work (Sheng et al., 2015, Soncin et al., 2018 and Chen and Ruedl 2020).

18. In Figure 3D, have they also analysed BAMs upon embryonic labelling?

We have analysed also BAMs at E7.5 embryonic labelling. We have included these data as new supplementary figure (Figure 4, figure supplement 2).

19. In Figure 4, MHC II+ BAMs seem to have a significantly higher turnover rate than MHC IIlow BAM. Do the authors have an explanation for that?

Currently we do not have any explanation for this differences. We have observed this higher turnover rate of MHCII^low^ tissue resident macrophages also in other tissue such as adipose tissue (Chen and Ruedl 2020) and exocrine pancreas (unpublised) and well as in intestinal tumors (Soncin et al., 2016). These MHCII^low^ tissue resident macrophages (often also Tim-4^+^) maintain their fetal origins and numbers via self-renewal. One could speculate that their niche is more tightly regulated than the MHCII^+^ resident macrophage niche, which is more “available” for monocyte replenishment. Interestingly these cells show higher YFP labelling in progeny of mice treated with TAM at E7.5. We can speculate that some of these MHC^low^ BAMs are descendants of yolk sac precursors.

20. The authors may discuss/consider that the slow turnover kinetics of BAM may be model dependent compared to other models where AD is accelerated.

It is not clear to us what the reviewer is referring too. Does the reviewer mean models where neuronal death occurs (tauTg models) or where there is more severe/accelerated time course of pathogenesis?

Since AD is a disease of aging, so severely accelerated pathogenesis may miss interactions between aging and AD pathogenesis in terms of turnover kinetics hence we believe that these model do not represent optimally the AD disease progression.

Other points that need to be addressed.21. In line 153, the authors might want to cite Jordão et al., 2019, who characterized a similar increase of MHC II+ CNS-associated macrophages in an EAE model with neuroinflammation.

We thank the reviewers for this citation which was now included in the result/Discussion section.

22. In line 172, it would be better to mention that these mice were harvested at 10 months of age.

This information was now included.

23. The MerCreMer labelling is not 100%, but around 80%. Is this a technical constraint that the authors should bring it up in their discussion as a caveat?

The original MerCreMer labelling, as other fate mapping models, will never be 100% (see attached picture in answer 17). In average, depending on the batch of mice and different days of treatment, we get a YFP labelling between 20-50% in neutrophils (see also figure attached to point 17 of this rebuttal). We consider neutrophils our reference cells due to their high turnover rate. Therefore we regard the labelling obtained in neutrophils as 100% (for each single mouse) and calculate accordingly the normalized labelling in % for each myeloid fraction in the brain of healthy and diseased animals.